# Use of Natural Products on the Control of *Aspergillus flavus* and Production of Aflatoxins In Vitro and on Tomato Fruit

**DOI:** 10.3390/plants10122553

**Published:** 2021-11-23

**Authors:** Mario Alberto Segura-Palacios, Zormy Nacary Correa-Pacheco, Maria Luisa Corona-Rangel, Ollin Celeste Martinez-Ramirez, Dolores Azucena Salazar-Piña, Margarita de Lorena Ramos-García, Silvia Bautista-Baños

**Affiliations:** 1Facultad de Nutrición, Universidad Autónoma del Estado de Morelos, Calle Iztaccihuatl S/N, Col. Los Volcanes, Cuernavaca C.P. 62350, Mexico; mario.ln9424@gmail.com (M.A.S.-P.); ollin.martinez@uaem.edu.mx (O.C.M.-R.); azucena.salazar@uaem.mx (D.A.S.-P.); 2Centro de Desarrollo de Productos Bióticos, Instituto Politécnico Nacional, Carretera Yautepec-Jojutla, km. 6.8, San Isidro, CEPROBI 8, Yautepec C.P. 62731, Mexico; zcorreap@ipn.mx (Z.N.C.-P.); mlcr7@hotmail.com (M.L.C.-R.)

**Keywords:** mycotoxins, coatings, nanoparticles, chitosan, citrus seed extract, pine resin extract

## Abstract

*Aspergillus flavus* affects fresh and dry fruit and vegetable products, and its toxic metabolites, namely aflatoxins, cause serious damage in humans. The objective of this research study was to evaluate the effect of commercial natural products as well as edible and nanostructured chitosan coatings on the development of *A. flavus* and on the production of aflatoxins in vitro and in tomato. Treatments were as follows: chitosan 1%, chitosan coating, chitosan nanostructured coating, Citrocover 1% (citrus seed extract), Resinadher 0.5% (pine resin extract), mancozeb 2%, and water. The variables were as follows: halo inhibition, spore production, and aflatoxins content. In fruit, the following were evaluated: disease incidence, mycelial growth, and aflatoxin production. An ANOVA (Tukey: *p* < 0.05) was used. In vitro results showed that Citrocover and Resinadher reduced sporulation (0.2 and 0.9 × 10^5^ spores mL^−1^, respectively), while chitosan inhibited the production of aflatoxins. With Resinadher and Citrocover, tomato fruit had the lowest incidence, mycelial growth, and aflatoxin production with corresponding values of 0%, 0.0 cm^2^, and 0.95 ppb, respectively, and 7%, 0.2 cm^2^, and 1.77 ppb, respectively. The use of Citrocover and Resinadher could be a viable alternative to decrease the development of *A. flavus* in tomato fruit.

## 1. Introduction

Tomato is a perishable fruit and can be contaminated by microorganisms mainly during storage. This is due to the high storage temperatures, increase in relative humidity, and increase in contact with fruit contaminated with microorganisms. The fungi *Alternaria alternata*, *Rhizopus stolonifer*, *Colletotrichum gloeosporiodes*, and *Aspergillus flavus* can contaminate tomatoes during storage, causing physical damage to the fruit. *Aspergillus flavus*, in addition to causing physical damage to the fruit, also produces toxic secondary metabolites for the human body called aflatoxins [1]. The consumption of aflatoxins has been confirmed to have carcinogenic, toxic, and mutagenic activity in humans and animals [2,3].

Many papers have reported the contamination of *A. flavus* with products such as corn, peanuts, pistachios, Brazil nuts, cotton seeds, grated dried coconut, sunflower seeds, soybeans, and nuts [4]. However, its presence in fresh fruit and vegetables has recently been reported. Buchanan et al. [5] carried out an investigation in figs contaminated by *A. flavus* and reported the presence of aflatoxins in all the ripening stages of the fruit, in which the concentrations were higher when the fruit was fully ripe (28 µg/g of fruit). Additionally, Baiyewu et al. [6] reported the presence of *A. flavus* and aflatoxins in papaya fruit, especially in those showing symptoms of microorganisms’ infection, while Maroutti [7] reported the identification of the fungi *A. flavus* and *A. parasiticus* as well as the presence of aflatoxins in fresh tomato fruit and by-products (pulp, paste, purée, ketchup, dehydrated tomatoes, and dried tomatoes preserved in oil).

Chemical fungicides have been mainly used to reduce the development of phytopathogenic fungi in tomatoes; however, some can be neurotoxic, genotoxic, teratogenic, and mutagenic, and can affect human health [8]. The application of these products, as well as the residues that remain in food, seriously affect the health of the people consuming them [9]. The collateral effects of pesticides have led to the need to evaluate new natural, biodegradable, and safe alternative applications for fresh tomato that can satisfactorily control the growth of the fungus *A. flavus* and hence avoid the development of aflatoxins. Among the control methods based on natural products with fungicidal properties to reduce contamination by these fungi [10,11,12,13,14,15], chitosan (solution, chitosan coating, and chitosan nanoparticles) and extracts of plant species (citrus and pine resin) have been used. However, there is little research on the control of *A. flavus* and its production of aflatoxins in tomato fruit. At present, there are commercial products based on citrus extract and pine resin on the market that have not been evaluated to control the development of *A. flavus* and could be an alternative for future application by producers of Morelos, México.

The objective of this research study was therefore to evaluate the effect of various natural compounds based on chitosan and plant extracts on the development of *A. flavus* in vitro and on tomato during a given storage period.

## 2. Results

### 2.1. In Vitro

In the in vitro studies, no significant statistical differences were observed between the treatments evaluated on day seven of incubation. However, the fungus treated with Citrocover showed statistical differences (*p* < 0.05) from the control (10.4, 4.2, and 3.5 cm^2^) after four, five, and six days of incubation, respectively (Table 1). Regarding the sporulation, the treatments that notably showed less sporulation (*p* < 0.05) were Citrocover and Resinadher (0.2 and 0.9 × 10^5^ spores mL^−1^, respectively; Table 2). The chitosan coating did not show aflatoxin production (0.0). Other treatments that presented low aflatoxin production values were Mancozeb, chitosan, and Citrocover (0.09, 0.10, and 0.13 ppb, respectively), while in the control, the aflatoxin production was 1.21 ppm (Table 2).

### 2.2. In Situ

The fruit that presented the lowest mycelial growth and incidence of *A. flavus* at the end of storage were those treated with Resinadher (0.0 cm and 0%, respectively) and Citrocover (0.2 cm and 7%, respectively). Both treatments were statistically different (*p* < 0.05) compared with the fruit treated with water (1.9 cm and 87%; Table 3). The remaining treatments showed a decrease in the fungal growth; however, they were statistically similar to the control.

The chemical treatment with Mancozeb was the one that showed the highest incidence (100%) of the disease at the end of the six days of storage. Both commercial products notably controlled the infection during the whole storage period of tomato (Resinadher = 1000; Citrocover = 95%; Figure 1).

In general, all the treated fruit showed a significant (*p* < 0.05) decrease in the production of aflatoxins compared to the untreated fruit. The fruit with the lowest values of aflatoxin production were those where Resinadher, Citrocover, and chitosan (0.95, 1.77, and 1.87, respectively; Table 4) were applied.

## 3. Discussion

### 3.1. In Vitro

The Citrocover treatment showed the greatest control on *A. flavus* during in vitro incubation. These results agree with Iglesias’ [16] findings. The author evaluated the extracts of in vitro studies of *Citrus aurantium*, *C. latifolia*, and *C. reticulata* on *Alternaria solani* and *Passalora fulva*, and reported that, in general, all extracts achieved a notable inhibition of the fungi greater than 50%. The citrus extracts are rich in bioactive compounds such as ascorbic acid, flavonoids, phenolic compounds, anthocyanins, pectins, alkaloids, glycosylases, saponins, steroids, and tannins, among others. Flavonoids and phenols are the main compounds that present a defense against phytopathogenic fungi [17]. Due to their aromatic structure and hydroxyl group, they alter the enzymatic activities of chitinase, chitin synthase, and β-glucanase of the cytoplasmic membrane of microorganisms [18].

In the Petri plates containing the chitosan coating, no production of aflatoxins (0.0) from the fungus *A. flavus* was recorded. Our data agree with the findings of Cortes-Higareda et al. [13]. These authors evaluated, in vitro, a chitosan coating at 1% and reported a complete inhibition of aflatoxin production. This could be due to the fact that chitosan is capable of adsorbing aflatoxin AfB1 when the positive charges of its amino group interact with the negative charges of the oxygen atoms of the aflatoxins [19]. Barkai-Golan [20] mentioned that the zinc present in the environment where the fungi develop stimulates the production of aflatoxins in *A. flavus* and *A. parasiticus*. Additionally, it has been reported that chitosan acts in the chelation of zinc; therefore, the addition of chitosan to the medium causes an inhibition in the production of the aflatoxins of the fungus [21]. Chitosan also presents a mechanism that involves ion chelation and for this, it requires the -OH and -O groups of the D-glucossamine residues as binders, as well as two or more amino groups of the same chain to join the same ion [22].

### 3.2. In Situ

In this study, the citrus seed extract and pine resin extract notably controlled the mycelial growth and incidence of *A. flavus* in tomato. On the same line, Gaviria-Hernández [23] evaluated the commercial extracts of *C. sinensis* and *C. grandis* (0.05%) on the development of *Colletotrichum gloeosporioides* and *C. acutatum* in blackberry and reported a 100% inhibition in both mycelial growth and fungal biomass with these extracts. Additionally, Silva [24] applied 1% citrus extracts to control *Penicillium* spp. in apple and reported a disease control of c.s. 99.5%. The effectiveness of citrus extracts against fungi is due to the presence of high concentrations of phenolic compounds and flavonoids such as flavones and flavone glycosides, which are enzyme inhibitors that affect the fungal respiration in the same way that they have the ability to form both alcohols and esters that inhibit the growth of hyphae as well as prevent the germination of spores.

Data provided by the manufacturer indicate that Citrocover is made up of ascorbic acid, palmitic acid, mannitose, glucose, glycerides, fatty acids, amino acids, citric bioflavonoids mixed with groups of amines, tocopherols, and orange terpenes. There are several researchers that indicate that orange terpenes have fungicidal activities. For example, Quintana-Obregón et al. [25] reported a 100% inhibition of the mycelial growth of *A. tenuissima*, but when they added 1% orange terpene, a 20% less fungal germination was obtained. The terpenes are compounds that reduce cell division abd inhibit oxygen consumption, mitochondrial respiration, oxidative forsphorylation, and DNA synthesis. The D-limonene specifically increases the fluidity of fungal membranes, leading to a high non-specific permeability of the membrane and loss of integrity, while the antifungal activity of citronellol is attributed to the inhibition of ergorestrol synthesis in the fungal membrane, which results in osmotic and metabolic instability [26]. To date, there are evaluations of commercial products that are citrus-based extracts which have been tested on fungi such as *Penicillium* spp., *C. gloeosporioides*, *Cladosporium* sp., *R. stolonifer*, *Lasiodiplodia theobromae*, *Phomopsis* sp., and *Alternaria* sp. Additionally, in other agricultural commodities such as apple, papaya, and passion fruit, some commercial products have shown a good control of these fungi [24,27]; however, others have shown limitations in its effect [28]. In the case of Citrocover, fungal growth inhibition, a decrease of sporulation, and a low production of aflatoxins were observed, therefore, it could be an alternative for future use as a biofungicide.

In the same way, the pine resin had an effect on the control of *A. flavus* development. This information agrees with that reported by Schultz [29]. The author evaluated the antimicrobial activity of *Pinus occidentalis* resin derivatives from a colophony resinous extract and turpentine essential oil. The colophony extract was effective against *Microsporum canes*, *Microsporum gypsium*, *Trichophyton rubum*, and *Pseudomonas aeruginosa*, while the essential oil of turpentine was effective against *M. canes*, *M. gypsium*, *Candida albicans*, and *P. aeruginosa*. The resin is a mixture of resin acids (60–75%), terpenes (10–15%), and other components such as alcohols and esters. Colophony and turpentine are the most common resins that are present in pine resin and are attributed to the ability of protecting against microorganisms [29,30]. The antimicrobial and antifungal activities of colophony have been attributed to the presence of hydroxyl groups, aldehyde, and ketones [31]. Turpentine has antiseptic properties; is rich in terpenes and terpenoids; is composed mainly of alpha pinene (87.1–94.1%), beta pinene, and 3 carene; and contains other terpenes such as limonene, pharesen, borneol, camphene, terinolene, and methylcarvinol. The reported antimicrobial activity suggests that the mechanism of action is through interactions between terpenes, altering the enzymatic system related to the energy production and synthesis of structural components within the microorganism [32].

The use of commercial products based on citrus seed extract (Citrocover) and pine resin extract (Resinadher) could be a viable alternative to reduce the development of *A. flavus* and its production of aflatoxins in tomato fruit.

## 4. Materials and Methods

### 4.1. Fungal Strain

The *A. flavus* fungus was obtained from the fungal collection of the Laboratory of Postharvest Technology of Agricultural Products at CEPROBI-IPN. The strain was activated on tomato fruit where mycelia and conidia were collected and incubated on Czapeck-dox agar medium at 20 °C.

### 4.2. Materials

The tomato Saladette type was obtained from an orchard located in Tetela del Volcán, México (10°53′35″ N 98°43′47″ O). The medium molecular weight of chitosan (Sigma-Aldrich, CAS: 9012-76-4; deacetylation degree of 75–85%) was used for the chitosan coating and nanoparticles synthesis. Glycerol was bought to J.T. Baker. The glacial acetic acid was purchased from Fermont Chemicals Inc. The citrus seed extract (Citrocover) and pine resine extract (Resinadher) were obtained from MS Agros S.A de C. V. Citrocover is *C. limon* and *C. sinensis*-based. The extraction of the plant material was carried out by steam distillation. Subsequently, the extracted material was mixed with inert diluents at a concentration of 7%. Resinadher products are pine resin-based and were obtained from *P. teocote*. The resin was collected manually and cleaned as well as later mixed with inert materials and diluents at a concentration of 60%.

### 4.3. Treatments Preparation

Seven treatments were evaluated: (1) chitosan 1%; (2) chitosan coating 1%; (3) chitosan nanoparticles 0.05%; (4) citrus seed extract (Citrocover) 1%; (5) pine resin extract (Resinadher) 0.5%; (6) chemical fungi (Mancozeb); and (7) water.

The chitosan concentration of 1.0% was prepared by adding the equal amount (*w*/*v* 1:100) of acetic acid to chitosan. The mixture was added to the total volume of distilled water and stirred overnight at room temperature. The solution was adjusted to pH 5.5 with 1N NaOH solution [13]. For the chitosan coating, 0.3% glycerol as a plasticizer was added to the previously formulated 1% chitosan solution. The chitosan nanoparticles were synthesized according to the methodology proposed by Correa-Pacheco et al. [33]. Chitosan of the medium molecular weight was used at a concentration of 0.05% (*w*/*v*) and dissolved in both glacial acetic acid (1% *v*/*v*) and distilled water. In total, 2.5 mL of this chitosan solution was dissolved in methanol (40 mL) using a peristaltic pump (Bio-Rad, EP-1 Econo Pump, USA) under moderate stirring. The obtained solution was placed in a rotary evaporator (Rotary Evaporator RE 300, BM 500 Water Bath, Yamato CF 300) at 40 °C and 50 rpm. The final volume of the nanoparticles was 2 mL. The zeta potential of the solution was +21.3 mV ± 0.06. The citrus seed and pine resin extracts as well as the chemical fungicide were diluted in distilled water using the concentrations recommended by the supplier (1%, 0.5%, and 2%, respectively).

### 4.4. Application of the Treatments In Vitro and on Tomatoe

In vitro: After Czapeck–dox culture medium solidification, 1.0 mL of *A. flavus* spore solution (10^5^ spore mL^−1^) was added and uniformly dispersed on Petri plates. A 0.5 cm hole was made in the center of the Petri dish and 20 µL of each treatment was added in the hole. Petri dishes were stored for 7 days at 28 + 2 °C. The inhibition halo of the fungus was measured every day in each treatment with a Truper vernier caliper during 7 days of incubation. Ten Petri plates were used per treatment. Fungal sporulation was evaluated at the end of the experiment using three randomly selected plates from each treatment. The conidia suspension was obtained by adding 10 mL of distilled water to the Petri dishes and the conidia concentration was calculated using a Neubauer chamber [34].

In situ: Tomato fruit were washed with 1% sodium hypochlorite and rinsed with sterile distilled water. The fruit were immersed in the treatments for 30 s and dried at room temperature. A wound was made in the fruit with a sterile dissection needle (2 mm deep) and 20 μL of a spore solution of *A. flavus* (10^5^ spores mL^−1^) was added; the fruit were stored at room temperature (28 °C ± 2 °C) for 6 days. Ten fruit were used per treatment [14]. The incidence (%), mycelial growth (cm), and aflatoxin production (ppb) of *A. flavus* in tomato were evaluated at the end of storage.

### 4.5. Aflatoxin Production by A. flavus

For the quantification of aflatoxins, the Raptor Neogen (USA) equipment and the Reveal Q+ for the aflatoxin kit were used. Reveal Q+ for aflatoxin is a single-step lateral flow immunoassay and the Raptor^®^ Integrated platform is a lateral flow test strip reader with built-in incubation.

In vitro: At the end of the storage, 10 mL of sterile distilled water was added to each Petri dish to obtain a solution. Additionally, 50 mL of 65% ethanol was added and mixed in an Oster (Mexico) food processor for 30 s at 10,000 RPM.; furthermore, 2 mL of this solution was taken and placed in Eppendorf tubes, as well as centrifuged at 12,500 RPM for 1 min. Subsequently, 100 µL of the supernatant was taken and 500 µL of developer liquid was added and mixed. Finally, 400 µL of this solution was taken and placed in the Neogen raptor equipment for reading.

In situ: at the end of the storage, the fruit were crushed. In total, 10 g of the sample was taken and 50 mL of ethanol was added; the procedure described in the previous paragraph was followed.

### 4.6. Statistical Analysis

The treatments were organized in a completely randomized design. Mean and standard deviations were calculated. The in vitro and in situ variables (mycelial growth, sporulation, and aflatoxin content) were subjected to ANOVA and a comparison of the means by Tukey’s test at *p* < 0.05 was conducted.

## 5. Conclusions

Citrus seed and pine resin extracts effectively inhibited the mycelial growth of *A. flavus* and aflatoxin production, while the chitosan coating reduced only the aflatoxin production. The commercial products Citrocover and Resinadher based on citrus seed and pine resin extracts could be viable alternatives for the control of the fungus *A. flavus* on tomato.

## Figures and Tables

**Figure 1 plants-10-02553-f001:**
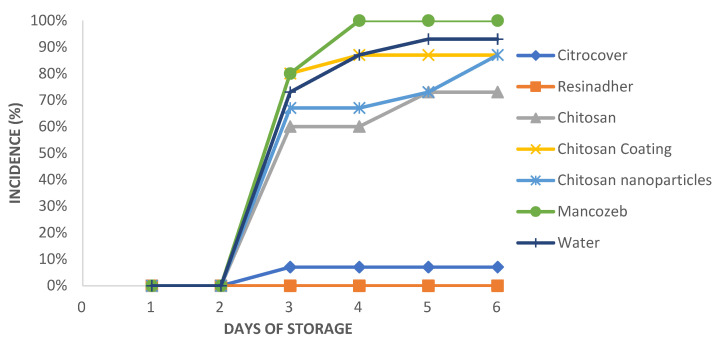
Incidence of *A. flavus* on tomato fruit treated with different natural products during six days of storage.

**Table 1 plants-10-02553-t001:** Inhibition halo of *A. flavus* treated with different natural products during seven days incubation in the in vitro study.

Treatment	Halo of Inhibition (cm^2^) (*x* ± S.D.)			
	Days of Incubation			
	4	5	6	7
Citrocover	10.4 ± 4.4 ^b^*	4.2 ± 5.1 ^b^	3.5 ± 4.7 ^b^	3.0 ± 4.5 ^a^
Resinadher	5.4 ± 5.0 ^ab^	2.6 ± 2.1 ^ab^	2.6 ± 2.4 ^ab^	2.6 ± 2.4 ^a^
Chitosan	0 ± 0 ^a^	0 ± 0 ^a^	0 ± 0 ^a^	0 ± 0 ^a^
Chitosan coating	0 ± 0	0 ± 0 ^a^	0 ± 0 ^a^	0 ± 0 ^a^
Chitosan nanoparticles	0 ± 0 ^a^	0 ± 0 ^a^	0 ± 0 ^a^	0 ± 0 ^a^
Mancozeb	2.6 ± 1.3 ^ab^	2.6 ± 1.5 ^ab^	1.1 ± 0.1 ^ab^	0.9 ± 0.15 ^a^
Water	0 ± 0 ^a^	0 ± 0 ^a^	0 ± 0 ^a^	0 ± 0 ^a^

* Means followed by the same letter are not significantly different (*p* < 0.05) determined by Tukey’s multiple test.

**Table 2 plants-10-02553-t002:** Sporulation and production of the aflatoxins of *A. flavus* treated with different natural products after seven days incubation in the in vitro study.

Treatment	Sporulation (10^5^ Spores mL^−1^)	Production of Aflatoxins (ppb)
Citrocover	0.2 ± 0 ^a^*	0.13 ± 0.16 ^a^
Resinadher	0.9± 0.02 ^a^	0.29 ± 0.25 ^ab^
Chitosan	4.7 ± 0.6 ^b^	0.10 ± 0.08 ^a^
Chitosan coating	14.5 ± 4.8 ^cd^	0.00 ± 0.00 ^a^
Chitosan nanoparticles	19.7 ± 4.1 ^d^	0.83 ± 2.20 ^ab^
Mancozeb	9.4 ± 3.1 ^bc^	0.09 ± 0.16 ^a^
Water	11.1 ± 1.14 ^c^	1.21 ± 1.50 ^b^

* Means followed by the same letter are not significantly different (*p* < 0.05) determined by Tukey’s multiple test.

**Table 3 plants-10-02553-t003:** Effect of natural products on the mycelial growth of *A. flavus* in tomato fruit during six days of storage.

Treatment	Mycelial Growth (cm^2^; *x* ± S.D.) Days of Storage				
	2	3	4	5	6
Citrocover	0.0 ± 0.0	0.04 ± 0.1 ^ab^*	0.07 ± 0.2 ^a^	0.1 ± 0.5 ^a^	0.2 ± 0.8 ^ab^
Resinadher	0.0 ± 0.0	0.0 ± 0.0 ^a^	0.0 ± 0.0 ^a^	0.0 ± 0.0 ^a^	0.0 ± 0.0 ^a^
Chitosan	0.0 ± 0.0	0.2 ± 0.2 ^abc^	0.7 ± 1.0 ^bc^	0.9 ± 1.2 ^abc^	1.0 ± 1.3 ^abc^
Chitosan coating	0.0 ± 0.0	0.2 ± 0.2 ^bc^	0.3 ± 0.2 ^abc^	0.4 ± 0.2 ^ab^	1.2 ± 0.9 ^abc^
Chitosan nanoparticles	0.0 ± 0.0	0.2 ± 0.3 ^bc^	0.3 ± 0.3 ^ab^	0.4 ± 0.4 ^a^	1.5 ± 1.5 ^bc^
Mancozeb	0.0 ± 0.0	0.3 ± 0.2 ^c^	0.0 ± 0.5 ^c^	1.3 ± 0.9 ^bc^	1.3 ± 1.9 ^bc^
Water	0.0 ± 0.0	0.1 ± 0.2 ^abc^	0.6 ± 0.5 ^bc^	1.8 ± 1.8 ^c^	1.9 ± 1.8 ^c^

* Means followed by the same letter are not significantly different (*p* < 0.05) determined by Tukey’s multiple test.

**Table 4 plants-10-02553-t004:** Effect of natural products in the production of the aflatoxins of *A. flavus*-treated tomato after six days of storage.

Treatment	Aflatoxins (ppb)
Citrocover	1.77 ± 0.21 ^a^*
Resinadher	0.95 ± 0.68 ^a^
Chitosan	2.12 ± 0.09 ^a^
Chitosan coating	2.50 ± 0.89 ^a^
Chitosan nanoparticles	1.87 ± 0.45 ^a^
Mancozeb	1.93 ± 0.65 ^a^
Water	5.00 ± 0.89 ^b^

* Means followed by the same letter are not significantly different (*p* < 0.05) determined by Tukey’s multiple test.

## Data Availability

Not applicable.

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
