# Peer review of "Use of Natural Products on the Control of Aspergillus flavus and Production of Aflatoxins In Vitro and on Tomato Fruit"

_plants, 2021, doi:10.3390/plants10122553_

Round 1

Reviewer 1 Report

This manuscripts describes the suppressive effects of several natural products on the mycelial growth of Aspergillus flavus and aflatoxin production in in vitro and on tomato fruits. The data may give interest to the readers. However, the paper was written poor English, therefore the paper should be edited by native English speaker before resubmission. Additionally, the conclusion and significance of the study should be provided in Abstract and Discussion sections.

Line 15: Aspergillus flavus affects ...........and its toxic metabolites, aflatoxins, cause serious damage in humans.

Line 23: in vitro results showed

Line 34: tomatoes during storage

Lines 34-35: Rewrite

Line 35: This last fungus --> A. flavus

Line 38: Rewrite to have carcinogenic, toxic and mutagenic activities in humans and animals.

Line 40: Rewrite. Many papers have reported the contamination

Line 51: Chemical fungicides have been mainly used to reduce

Lines 51-54; Rewrite

Line 71: in vitro = italic

Line 80: A. flavus = italic

Line 81: delete comma(,)

Line 82: ). Both treatments

Line 89: the fruit were

Lines 91 & 96: delete comma

Lines 100&101: A. flavus = italic

Lines 110 and 111: Rewrite

In the first paragraph of Discussion section, the authors had better describe the general background of the study.

Lines 111-112: Rewrite.

Lines 119-123: Rewrite.

Line 125: in in in vitro ?????

Line 147: “compromising its reproduction” What does it mean?

In the Abstract and Discussion sections, the authors should provide the significance of the study.

Line 181: Provide the variety if tomato plants

Lines 197-200: Rewrite

Line 207: 1.0 mL?? The volume is so big.

Lines 216-217: Rewrite.

Line 217: How did you make wounds on tomato fruit?

Line 220: Ten fruit

What is the replication? Did you repeat the experiment?

Conclusions: It should be revised. Both citrus seed and pineapple resin extracts effectively inhibited the mycelial growth of A. flavus and aflatoxin production, but the chitosan coating reduced only the aflatoxin production without any effect on the mycelial growth. Therefore, the ............................

Author Response

REVIEWER 1

Comment: This manuscript describes the suppressive effects of several natural products on the mycelial growth of Aspergillus flavus and aflatoxin production in in vitro and on tomato fruits. The data may give interest to the readers. However, the paper was written poor English, therefore the paper should be edited by native English speaker before resubmission. Additionally, the conclusion and significance of the study should be provided in Abstract and Discussion sections.

Answer: The manuscript was sent to the English revision. I attach proof of review.

Comment: Line 15: Aspergillus flavus affects ...........and its toxic metabolites, aflatoxins, cause serious damage in humans.

Answer: Correction was made in the text

Comment: Line 23: in vitro results showed

Answer: Correction was made in the text

Comment: Line 34: tomatoes during storage

Answer: Correction was made in the text

Comment: Lines 34-35: Rewrite

Answer: Correction was made in the text.The fungi Alternaria alternata, Rhizopus stolonifer, Colletotrichum gloeosporiodes and Aspergillus flavus can contaminate tomatoes during storage causing physical damage to the fruit

Comment: Line 35: This last fungus --> A. flavus

Answer: Correction was made in the text

Comment: Line 38: Rewrite to have carcinogenic, toxic and mutagenic activities in humans and animals.

Answer: Correction was made in the text

Comment: Line 40: Rewrite. Many papers have reported the contamination

Answer: Correction was made in the text

Comment: Line 51: Chemical fungicides have been mainly used to reduce

Answer: Correction was made in the text

Comment: Lines 51-54; Rewrite

Answer: Correction was made in the text.Chemical fungicides have been mainly used to reduce the development of phytopathogenic fungi in tomatoes, however some can be neurotoxic, genotoxic, teratogenic, mutagenic and can affect human health [8].”

Comment: Line 71: in vitro = italic

Answer: Correction was made in the text

Comment: Line 80: A. flavus = italic

Answer: Correction was made in the text

Comment: Line 81: delete comma(,)

Answer: Correction was made in the text

Comment: Line 82: ). Both treatments

Answer: Correction was made in the text

Comment: Line 89: the fruit were

Answer: Correction was made in the text

Comment: Lines 91 & 96: delete comma

Answer: Correction was made in the text

Comment: Lines 100&101: A. flavus = italic

Answer: Correction was made in the text

Comment: Lines 110 and 111: Rewrite

Answer: Correction was made in the text.In the petri dishes containing the chitosan coating, no production of aflatoxins (0.0) from the fungus A. flavus was observed.”

Comment: In the first paragraph of Discussion section, the authors had better describe the general background of the study.

Answer: Correction was made in the text

Comment: Lines 111-112: Rewrite.

Answer: Correction was made in the text. 'These data agree with Cortes-Higareda et al. [13]. The authors evaluated a chitosan coating at 1% in an in vitro study and reported a complete inhibition of aflatoxin production'.

Comment: Lines 119-123: Rewrite.

Answer: Correction was made in the text. “Chitosan presents a mechanism that involves ion chelation and for this, it requires the -OH and -O groups of the D-glucossamine residues as binders and 2 or more amino groups of the same chain to join the same ion [19].”

Comment: Line 125: in in in vitro ?????

Answer: Correction was made in the text

Comment: Line 147: “compromising its reproduction” What does it mean?

Answer: Correction was made in the text. The text that generated confusion was deleted.

Comment: In the Abstract and Discussion sections, the authors should provide the significance of the study.

Answer: Correction was made in the text

Comment: Line 181: Provide the variety if tomato plants

Answer: Correction was made in the text

Comment: Lines 197-200: Rewrite

Answer: The paragraph was modified without modifying the processes carried out

Comment: Line 207: 1.0 mL?? The volume is so big.

Answer: The data is correct. 1mL was used, because the spore solution was spread throughout the petri dish

Comment: Lines 216-217: Rewrite.

Answer: Correction was made in the text. “Tomato fruit were washed with 1% sodium hypochlorite and rinsed with sterile distilled water.”

Comment: Line 217: How did you make wounds on tomato fruit?

Answer: Correction was made in the text

Comment: Line 220: Ten fruit

Answer: Correction was made in the text

Comment: What is the replication? Did you repeat the experiment?

Answer: The experiment was repeated

Comment: Conclusions: It should be revised. Both citrus seed and pineapple resin extracts effectively inhibited the mycelial growth of A. flavus and aflatoxin production, but the chitosan coating reduced only the aflatoxin production without any effect on the mycelial growth. Therefore, the….

Answer: Both citrus seed extract and pine resin extract effectively inhibited the mycelial growth of A. flavus and aflatoxin production, but the chitosan coating reduced only aflatoxin production without any effect on mycelial growth. The commercial products Citrocover and Resinadher based on citrus seed and pine resin extracts could be a viable alternative for the control of the fungus A. flavus on tomato.

Reviewer 2 Report

The article entitled "Use of Natural Products on the Control of Aspergillus flavus and Production of Aflatoxins In Vitro and On Tomato Fruit" Needs to address following comments and questions. 

1. in the results part, there are confusing sentences and hard to understand the conclusion. In line 71, the authors mentioned that" In the in vitro studies, no significant statistical differences were observed between 71 the evaluated treatments, while in Line 86 they mentioned " In general, all the treated fruit  showed a decrease in the production of aflatoxins and were statistically different from the control. 

2. In Line 110: "In this study, the chitosan coating did not show aflatoxin production (0.0) on A. flavus. While in the table 4 they reported 2.5 ppb aflatoxins for chitosan test?

3. What was the cationic charge density of applied chitosan? 

4. What was the antifungal effect of chitosan and extract materials

5. How citrus, pine resin and other extractives were extracted? What was the procedure? What type of active materials were extracted? Without knowing this information, the results are not meaningful. 

Author Response

REVIEWER 2

Comment: The article entitled "Use of Natural Products on the Control of Aspergillus flavus and Production of Aflatoxins In Vitro and On Tomato Fruit" Needs to address following comments and questions. 

Comment: 1. in the results part, there are confusing sentences and hard to understand the conclusion. In line 71, the authors mentioned that" In the in vitro studies, no significant statistical differences were observed between 71 the evaluated treatments, while in Line 86 they mentioned " In general, all the treated fruit  showed a decrease in the production of aflatoxins and were statistically different from the control. 

Answer: Two evaluations are carried out in vitro and in situ, the first paragraph refers to the in vitro study (it is specified in the text of the manuscript), while the remaining information refers to the in situ study.

Comment: 2. In Line 110: "In this study, the chitosan coating did not show aflatoxin production (0.0) on A. flavus. While in the table 4 they reported 2.5 ppb aflatoxins for chitosan test?

Answer: The paragraph refers to in vitro studies while in it the data are in tomato fruit

Comment: 3. What was the cationic charge density of applied chitosan? 

Answer:  It is not clear this  question. Is it about the degree of deacetylation? If that is so, the information is in the methodology.

Comment: 4. What was the antifungal effect of chitosan and extract materials? 

Answer: The antifungal effect of chitosan and citrus extracts was placed in the conclusion

Comment: 5. How citrus, pine resin and other extractives were extracted? What was the procedure? What type of active materials were extracted? Without knowing this information, the results are not meaningful. 

Answer: The vegetable extracts of citrus (Citrocover) and pine resin (Resinadher) are commercial products.

Round 2

Reviewer 1 Report

The authors addressed well the comments raised by me.

Author Response

Thanks to the reviewer, for the comments.

Reviewer 2 Report

Comment: 1. in the results part, there are confusing sentences and hard to understand the conclusion. In line 71, the authors mentioned that" In the in vitro studies, no significant statistical differences were observed between 71 the evaluated treatments, while in Line 86 they mentioned " In general, all the treated fruit showed a decrease in the production of aflatoxins and were statistically different from the control.

Answer: Two evaluations are carried out in vitro and in situ, the first paragraph refers to the in vitro study (it is specified in the text of the manuscript), while the remaining information refers to the in situ study.

This is the confusing part of the article. Invitro and in-situ data  are mixed together. please separate each part and discuss individually  and compare them at the end. 

Comment: 2. In Line 110: "In this study, the chitosan coating did not show aflatoxin production (0.0) on A. flavus. While in the table 4 they reported 2.5 ppb aflatoxins for chitosan test?

Answer: The paragraph refers to in vitro studies while in it the data are in tomato fruit

Same issue, the discussion is not matched with table and graph data. 

Comment: 3. What was the cationic charge density of applied chitosan?

Answer: It is not clear this question. Is it about the degree of deacetylation? If that is so, the information is in the methodology.

Chitosan after acid treatment has cationic charge ( + ), depending on the type of chitosan and pretreatment, the density of the charge will be different and has a big effect on its biological activity. easy to measure by zeta-potential  

Comment: 4. What was the antifungal effect of chitosan and extract materials?

Answer: The antifungal effect of chitosan and citrus extracts was placed in the conclusion

Can't see any data

Comment: 5. How citrus, pine resin and other extractives were extracted? What was the procedure? What type of active materials were extracted? Without knowing this information, the results are not meaningful.

Answer: The vegetable extracts of citrus (Citrocover) and pine resin (Resinadher) are commercial products.

These are commercial products, what is the product chemical specification? We can't use a commercial product without knowing their chemical spec. in a scientific article and make a conclusion about their effect. Need more information to be repeatable by others. Depending on the citrus and pine species and the method of oil extraction, you can find wide range of materials with different activities from same plant. Need to know the plant species and extraction method or have a chemical analysis before use. 

Author Response

Comment: 1. in the results part, there are confusing sentences and hard to understand the conclusion. In line 71, the authors mentioned that" In the in vitro studies, no significant statistical differences were observed between 71 the evaluated treatments, while in Line 86 they mentioned " In general, all the treated fruit showed a decrease in the production of aflatoxins and were statistically different from the control.

Answer: In this study, two different experiments were carried out, in vitro and in situ. The information was divided in two paragraphs, they were labeles as: 'In vitro' nd 'In situ'

This is the confusing part of the article. In vitro and in-situ data are mixed together. please separate each part and discuss individually and compare them at the end. 

Answer: To avoid confusion, the two methods were labelled separately and explained in the sections of Results, Discussion and Material end methods the two

Comment: 2. In Line 110: "In this study, the chitosan coating did not show aflatoxin production (0.0) on A. flavus. While in the table 4 they reported 2.5 ppb aflatoxins for chitosan test?

Answer: This information refers to the in vitro results. The value of 2.5 ppb is for the analysis of aflatoxins carried on tomato.

Same issue, the discussion is not matched with table and graph data. 

Answer: To avoid confusion, this section was divided in 'In vitro' and 'In situ', the interpretation was given then separately.

Comment: 3. What was the cationic charge density of applied chitosan?

Answer: The degree of deacetylation is given in the methodology.

Chitosan after acid treatment has cationic charge ( + ), depending on the type of chitosan and pretreatment, the density of the charge will be different and has a big effect on its biological activity. easy to measure by zeta-potential  

Answer:  The potential zeta of the solution was included in the methodology.  “The zeta potential of the solution was + 21.3mV ± 0.06”

Comment: 4. What was the antifungal effect of chitosan and extract materials? Can't see any data.

Answer: In Figure 2, it is shown the results of this variable. Discussion was a little extended, pointing out the main results of the disease incidence.

Comment: 5. How citrus, pine resin and other extractives were extracted? What was the procedure? What type of active materials were extracted? Without knowing this information, the results are not meaningful.

Answer: The vegetable extracts of citrus (Citrocover) and pine resin (Resinadher) are both commercial products.

These are commercial products, what is the product chemical specification? We can't use a commercial product without knowing their chemical spec. in a scientific article and make a conclusion about their effect. Need more information to be repeatable by others. Depending on the citrus and pine species and the method of oil extraction, you can find wide range of materials with different activities from same plant. Need to know the plant species and extraction method or have a chemical analysis before use. 

Answer: It is true, but due to the pandemic situation, the laboratory that carries out the chemical profile of natural products in the University is closed. However, we attempt to discuss these results with previous published information about the fungicidal activity of the Citrus and Pine genus. Definitely, as soon as the time allows, that kind of analysis will be perform.

Information about the extraction methods was added in the section of Materials and methods.

Round 3

Reviewer 2 Report

Thank you for all of your effort to  address all of the questions and comments. Please read the article one more time and try to remove any confusion between in situ and in vitro discussion. please change table 1 and 2 tilts as follow

Change table 1 and 2 titles as:

  1. Table 1. Inhibition halo of A. flavus treated with different natural products during seven days incubation in the in vitro study
  2. Table 2. Sporulation and production of aflatoxins of A. flavus treated with different natural products after seven days incubation in the in-situ study